# Serum Anti-Müllerian Hormone Levels and Estrous Monitoring of GnRH Agonist Deslorelin-Induced Estrus in Bitches: A Pilot Study

**DOI:** 10.3390/ani13020258

**Published:** 2023-01-12

**Authors:** Sroisuda Chotimanukul, Sandra Goericke-Pesch, Junpen Suwimonteerabutr, Jinda Singlor, Ekkaphot Sangkrachang, Padet Tummaruk, Suppawiwat Ponglowhapan

**Affiliations:** 1Department of Obstetrics Gynaecology and Reproduction, Faculty of Veterinary Science, Chulalongkorn University, Bangkok 10330, Thailand; 2Research Unit for Obstetrics and Reproduction in Animals, Chulalongkorn University, Bangkok 10330, Thailand; 3Reproductive Unit, Clinic for Small Animals, University of Veterinary Medicine Hannover, Foundation, 30559 Hannover, Germany

**Keywords:** AMH, deslorelin, dog, estrus induction, GnRH agonist

## Abstract

**Simple Summary:**

Deslorelin is a slow-release gonadotropin-releasing hormone (GnRH) agonist that has been successfully used to induce an estrous cycle in bitches. Little is known regarding the details of ovarian follicle development in estrous bitches induced by the deslorelin implant. Anti-Müllerian hormone (AMH), produced by granulosa cells in the ovary, is involved in the mechanism of follicle development. Nevertheless, information of AMH as a fertility marker in the bitch is scarce. The current study aimed to monitor estrous pattern and follicular development in deslorelin-induced estrous bitches. The changes in AMH concentrations and ovarian ultrasonography during the peri-ovulatory period were mainly investigated. Results showed that a slow-release deslorelin implant successfully induced estrus and ovulation in all bitches. The induction of estrus occurs shortly after deslorelin implantation. AMH concentrations varied greatly among bitches. No significant differences in AMH levels were observed between pre- and post-implantation. Concentrations of AMH and the pattern of AMH were different from natural estrus. Ovarian ultrasound revealed changes in follicular development over the observation period. It seemed that the fertile period of deslorelin-treated bitches was shorter than that of natural-occurring estrous bitches. In conclusion, our findings suggest that AMH is not suitable as a fertility marker to monitor ovarian response to deslorelin treatment for estrus induction in bitches.

**Abstract:**

This study was performed to monitor estrous patterns and, more importantly, changes in anti-Müllerian hormone (AMH) concentrations during the peri-ovulatory period in deslorelin-induced estrous bitches. Healthy anestrous bitches (*n* = 4) were used. Estrus and ovulation were monitored after deslorelin implantation. Blood samples were collected for analysis of progesterone, estradiol-17ß and AMH concentrations before implantation (day 0) and on days 6, 8, 10, 12, 14, 16, 18, 20 and 22 after implantation. Six days following treatment, all bitches showed estrus signs. Ovulation took place between days 12 and 15. Circulating AMH concentrations varied among bitches from 0.12 to 3.08 ng/mL. However, no significant differences in AMH levels (mean ± SD) were observed between day 0 and days following post-implantation *(p >* 0.05). There were no significant correlations between AMH and estradiol or AMH and progesterone *(p >* 0.05). Ultrasonographically, the number of clearly identifiable ovarian follicles was higher before ovulation and the area of ovaries increased after ovulation *(p <* 0.05). Except for AMH, changes in vaginal cytology, estradiol-17ß and progesterone levels observed in our study were similar to naturally occurring estrus. Large intra- and inter-individual variation in AMH were observed suggesting that AMH is currently not suitable as a canine fertility marker to monitor ovarian response to deslorelin treatment for estrus induction.

## 1. Introduction

Gonadotropin-releasing hormone (GnRH) agonist implants containing deslorelin have been used for estrus induction and suppression in bitches for decades [1,2,3,4,5,6]. In terms of estrus induction in the dog, deslorelin implants have been successfully applied using the flare-up effect [4,5,7,8,9,10,11]. Approximately 4 days (range 2–7 days) post-implantation, anestrus bitches come into estrus. Removal of the implant on the day of ovulation or shortly afterwards is recommended to avoid or at least reduce the risk for pituitary downregulation and subsequent secondary luteal failure, reported as a severe adverse effect from the prolonged administration of GnRH agonist implants [4]. Even though the results of estrus induction using deslorelin implants are satisfactory, ovulation failure has been described [4,8] and pregnancy rates vary among studies [4,9,10]. In general, the ovulation rate appears to be higher when dogs are induced in late anestrus [10,12,13].

Anti-Müllerian hormone (AMH) or Müllerian-Inhibiting Substance (MIS) is a dimeric glycoprotein of the transforming growth factor-β superfamily [14,15]. Although AMH is associated with sexual development of the male fetus, AMH is also produced during adulthood in testes, namely Sertoli cells, and ovaries, namely granulosa cells [16,17]. Research on AMH indicates that this hormone plays an important role during follicular development and function [14,18]. In women, AMH is one of the predictors for the ovarian reserve, ovarian response to hormonal treatment during in vitro fertilization (IVF) protocols and live birth rates and, recently, for the beginning of menopause [19,20,21]. In domestic animals, AMH is described to be a predictor of fertility, superovulation and ovarian disorders [18]. Despite being a valuable tool in other species, current knowledge on the clinical potential of AMH in the female dog is limited. The applicability of AMH has gained increasing attention in canine reproduction in recent years. Most studies in the bitch have been conducted to investigate the use of AMH as a diagnostic tool for reproductive pathologies, such as ovarian remnant syndrome, ovarian cysts and granulosa cell tumors [22]. In addition, a previous study found a positive correlation between AMH concentrations and litter size in bitches. Thus, AMH might be used as a predictor of canine litter size [23]. More recently, AMH has been reported as a useful tool to identify testicular tissue in dogs with disorders of sexual development, especially in phenotypic females [24]. However, information of AMH as a fertility marker in the bitch is scarce. Little is known regarding the details on follicular development in estrous bitches induced by GnRH agonist implants containing deslorelin. Therefore, the present study aimed to monitor estrus and follicular development in deslorelin-induced estrous bitches by assessing AMH concentrations during the peri-ovulatory period.

## 2. Materials and Methods

All experimental procedures were conducted in accordance with the ethical standards issued by the National Research Council and were approved by the Ethics Committee for Human and/or Animal Experimentation in meeting No. 1431070, Faculty of Veterinary Science, Chulalongkorn University, Thailand.

### 2.1. Animals

Healthy, nulliparous, anestrous, 2.5–4-year-old Beagle bitches weighing between 10.5 and 13.0 kg were enrolled in this study (*n* = 4). The bitches were considered in anestrus when their serum progesterone levels were below 1.0 ng/mL and the percentage of superficial cells from vaginal cytology was less than 10% [9].

### 2.2. Estrus Induction

A 4.7 mg deslorelin implant (Suprelorin^®^, Virbac, Carros, France) was inserted subcutaneously in the postumbilical area in all anestrous bitches on the same day. Time of implant injection was considered as day 0, and time from day 0 to estrus induction (first day of bleeding; visible vaginal bloody discharge) was calculated. The implant was removed at ovulation [4,5] when progesterone concentrations were more than 5.0 ng/mL, as determined using FEIA, fluorescence enzyme immunoassay [25].

### 2.3. Cytological Examination

Exfoliative vaginal cytology was performed on day 0 and days 6, 8, 10, 12, 14, 16 and 18 post-implantation. Vaginal cytology samples were obtained by introducing a saline-moistened cotton swab into the vagina and rotating it on the vaginal mucosa. The swab was rolled over a clean microscopic slide and stained using Diff Quick^®^ stain (CLINAG Co., Ltd., Bangkok, Thailand). The slides were examined using a light microscope (Olympus BX51, Olympus, Tokyo, Japan) at 400× magnification. Epithelial cells were counted as superficial or non-superficial cells in five random areas and the percentages of superficial cells were calculated.

### 2.4. Hormone Analysis

Blood samples were collected on days 0, 6, 8, 10, 12, 14, 16, 18, 20 and 22 after implant injection from the cephalic vein into VACUETTE^®^ tubes serum clot activator (Greiner Bio-One (Thailand) Ltd., Chonburi, Thailand). Sera were obtained via blood centrifugation for 10 min at 700× *g* for AMH, estradiol and progesterone assays. Hormone measurements were conducted in duplicate.

#### 2.4.1. AMH Assay

AMH concentrations were determined by Canine AMH ELISA (AL-116, Ansh Labs, Webster, TX, USA) according to the manufacturer’s instructions [23,26]. Briefly, calibrators, controls and unknown samples were added to AMH antibody-coated micro-titer wells and incubated. After the first incubation and washing, the wells were incubated with biotinylated AMH antibody solution. After second incubation and washing, the wells were incubated with streptavidin horseradish peroxidase conjugate solution. After the third incubation and washing step, the wells were incubated with substrate solution followed by an acidic stopping solution. The lower limit of detection of the recent canine AMH ELISA assay was 0.055 ng/mL. The intra-assay coefficient of variation for the AMH ELISA was 5.3%. In this study, the inter-assay coefficient of variation for the AMH ELISA was not available because all samples were run in one batch.

#### 2.4.2. Estradiol-17ß Assay

Estradiol concentrations were determined by CMIA, chemiluminescent microparticle immunoassay (ARCHITECT Estradiol, Abbott Laboratories, Abbott Park, IL, USA) according to the manufacturer’s instructions [27]. Briefly, sample, specimen diluent, assay diluent and anti-estradiol (rabbit, monoclonal)-coated paramagnetic microparticles were combined. Estradiol present in the sample bound to the anti-estradiol-coated microparticles. After incubation, estradiol acridinium-labeled conjugate was added to the reaction mixture. After further incubation and washing, pre-trigger and trigger solutions were added to the reaction mixture. The resulting chemiluminescent reaction was measured as relative light units (RLUs). There was an inverse relationship between the amount of estradiol in the sample and the RLUs detected by the ARCHITECT iSystem optics. The assay precision was ≤5 pg/mL. The analytical sensitivity was ≤10 pg/mL. The intra- and inter-assay variabilities of the test were 5.5% and 6.7%, respectively.

#### 2.4.3. Progesterone Assay

Progesterone concentrations were determined by FEIA [25] (ST AIA-PACK PROGII, TOSOH Corp., Tokyo, Japan) according to the manufacturer’s instructions. Briefly, progesterone in the sample competed with enzyme-labeled progesterone for a limited number of binding sites on a progesterone-specific antibody immobilized on magnetic beads. The beads were washed to remove the unbound enzyme-labeled progesterone and were then incubated with a fluorogenic substrate, 4-methylumbelliferyl phosphate. The amount of enzyme-labeled progesterone that binds to the beads was inversely proportional to the progesterone concentration in the sample. A standard curve using a range of known standard concentrations was constructed and unknown progesterone concentrations were calculated using the resulting curve. The limit of detection was 0.1 ng/mL; the maximum, the highest measurable concentration, was 40 ng/mL. The intra- and inter-assay variabilities of the test were 3.4% and 4.1%, respectively.

### 2.5. Ultrasonographic Examination of Ovaries

Abdominal ultrasonography was performed for visualization of the ovaries using a Mindray M9 portable color doppler ultrasound machine (Shenzhen A-Faith Technology Co., Ltd., Shenzhen, China) equipped with a high-resolution micro-convex transducer. The ultrasound frequency remained at 8 MHz throughout B-mode scanning. The gain was adjusted for each scan in order to obtain optimal images. Dogs were positioned as previously described [28]. Abdominal hair was clipped in the areas corresponding to the approximate location of the ovaries in order to optimize ultrasound images. The transducer was swept across the ovaries in order to obtain clear images of the entire ovary [29]. The number of fluid-filled structures was counted separately for each ovary. Additionally, ovarian width and length on the largest cut surface were measured for each ovary to calculate the ovarian area by using the equation of an ellipse A = ∏ × a/2 × b/2 (a = width, b = length). Ultrasound examinations were performed on day 0 and days 6, 8, 10, 12, 14, 16 and 18 post-implantation by the same ultrasonographer.

### 2.6. Statistical Analysis

Data analyses were carried out using general linear mixed models. The circulating concentrations of AMH, estradiol and progesterone were regarded as dependent variables. The statistical models included day of ovulation as a fixed effect and bitch identity as a random effect. Least square means were obtained from each class of the factor and were compared by using least significant difference (LSD) test. Differences in hormone concentrations (mean ± SD) between observation days (day 0 and days 6, 8, 10, 12, 14, 16, 18, 20 and 22) were compared. The correlation between AMH and estradiol or progesterone levels was determined by using Pearson’s correlation. *p* < 0.05 was regarded to be statistically significant. The number of fluid-filled structures and area of the ovary (*n* = 8) were calculated using mean ± SD on individual days. Multiple analysis of variance using SAS version 9.4 (SAS Inst. Inc., Cary, NC, USA) was used to compare the differences in the number of fluid-filled structures and the ovarian area on individual days. Differences with *p* < 0.05 were regarded as statistically significant.

## 3. Results

### 3.1. Vaginal Cytology

Bitches showed signs of estrus between days 5 and 6 following implantation. The number of superficial cells increased sharply on day 6 after implant injection concomitantly with serum estradiol concentrations. The percentage of superficial cells in relation to serum estradiol concentration is shown in Figure 1. The high percentage of superficial cells (≥80%) was maintained constantly until day 14 (*n* = 3) and 16 (*n* = 1) (Figure 1). Four days after ovulation, the percentage of superficial cells decreased sharply to approximately 10%.

### 3.2. Hormone Analysis

Serum AMH, estradiol and progesterone concentrations were measured in a total of 40 samples from four dogs. Due to the limited number of dogs used in this study, the hormone concentrations are presented individually in Figure 2. Before implantation, the concentrations of AMH varied among anestrous bitches between 0.46 and 3.44 ng/mL. After implantation, ranges of AMH concentrations were between 0.12 and 3.08 ng/mL. No differences in AMH concentrations between day 0 (1.13 ± 1.27) and day 6 (1.21 ± 0.94), day 8 (0.97 ± 0.69), day 10 (1.04 ± 0.79), day 12 (0.77 ± 0.63), day 14 (1.23 ± 0.99), day 16 (1.15 ± 0.92), day 18 (1.68 ± 1.14), day 20 (1.22 ± 0.56) and day 22 (1.26 ± 0.99) were observed (all; *p* > 0.05). There was no significant correlation between AMH and estradiol or progesterone, respectively (all; *p* > 0.05).

Estradiol concentrations were lower than 10 pg/mL in all bitches on day 0 and the concentrations varied between 9.9 and 39.4 pg/mL post-implantation. The highest concentration in each dog was observed before ovulation between days 8 and 12. Estradiol concentrations tended to decrease sharply after reaching peak concentration. Mean estradiol concentrations were significantly higher on day 8 (28.25 ± 12.07) and day 10 (27.85 ± 11.98) compared to day 6 (17.97 ± 5.43), day 12 (15.57 ± 11.15), day 14 (11.40 ± 1.89), day 16 (10.55 ± 0.80), day 18 (10.00 ± 0.00), day 20 (10.00 ± 0.00) and day 22 (10.00 ± 0.00) (all; *p* < 0.05).

During the observation period, progesterone concentrations varied between below 0.1 and 25.64 ng/mL. Progesterone levels post-implantation increased significantly over time in all bitches, indicating successful estrus induction and induction of ovulation. The ovulation was observed between days 12 and 15 post-implantation (Figure 2). A significant difference in progesterone concentrations between day 0 (0.05 ± 0.06) and day 14 (7.07 ± 4.36), day 16 (17.78 ± 5.94), day 18 (17.86 ± 2.53), day 20 (17.66 ± 4.62) and day 22 (18.18 ± 5.31) was identified (all; *p* < 0.01).

### 3.3. Ovarian Ultrasound

Using ultrasound examination, changes in ovarian echogenicity of the follicles were noticed on day 6; however, identifiable fluid-filled structures were first clearly noticed on day 8 (Figure 3). The mean (±SD) number of clearly identifiable fluid-filled structures in both ovaries on individual days ranged from 3.5 ± 3.12 to 7.25 ± 0.96. Additionally, the greatest number of follicles was observed on day 8 compared with days 10, 12, 14 and 18 (*p* < 0.05) (Table 1). The mean (±SD) area of the ovaries on individual days ranged from 0.8 ± 0.06 to 2.41 ± 0.93 cm^2^. After ovulation (day 16), the ovarian area was significantly greater than before ovulation (days 8, 10, 12 and 14) (*p* < 0.05). Nevertheless, no significant difference was observed between days 16 and 18 (Table 1).

## 4. Discussion

Deslorelin-containing GnRH agonist implants successfully induced proestrus in all bitches, as indicated by changes in exfoliative vaginal cytology, serum estradiol concentrations and ovarian follicular growth. Ovulation was observed between days 12 and 15 of implantation, in accordance with a previous study [30]. Detection of ovulation by progesterone concentrations in bitches has been extensively used in clinical practice. The initial rise in serum progesterone concentrations more than 2 ng/mL indicates the occurrence of the luteinizing hormone (LH) surge and the ovulation usually occurs 2–3 days after the LH surge in estrous bitches [31,32]. This phenomenon was also revealed in deslorelin-induced estrous bitches in the present study, similarly to natural-occurring canine estrus. Serum progesterone concentrations dramatically increased from 2 ng/mL to reaching above 5 ng/mL within two days in all four bitches. It is worth noting that, four days post-ovulation, the percentage of superficial cells declined to approximately 10%, indicating the onset of cytologic diestrus. In the natural-occurring canine estrous cycle, the change in vaginal smears that signifies the onset of diestrus usually occurs about 5–7 days after ovulation [33]. In this study, it seems that the onset of cytologic diestrus occurred earlier than after naturally occurring estrus, thus, suggesting a shorter fertile window in deslorelin-induced estrus. Differences in fertile window after ovulation between untreated and deslorelin-induced estrus cycles in a larger number of bitches warrant further investigation.

Different to earlier studies [23,26,34,35], overall AMH levels were much lower in the present study. It was intriguing to note that earlier studies used either human or canine ELISA AMH kits to detect AMH concentrations in dogs; however, two earlier studies used the same ELISA AMH kit with a higher limit of detection (0.2 ng/mL) [23,26]. In one of those studies [26], some intact dogs had serum AMH levels below the detection limit, so new standard curve points were implemented in the Canine AMH ELISA (AL-116, Ansh Labs, Webster, TX, USA) used in our study to identify lower AMH concentrations (limit of detection: 0.055 ng/mL) (personal communication with Ajay Kumar from the manufacturer). This might explain some of the observed differences between our and previous findings. In this study, one dog (number 4, Figure 2) had higher AMH concentrations (3.08 ng/mL) on day 0 compared to other dogs, suggesting a large variation in AMH levels between anestrous bitches. It was possible that the ovarian cycle of this dog (number 4) in our study was in the course of the anestrous to oestrous transition because a significant increase in AMH concentrations is reported in this stage of canine ovarian cycle [14], and the highest AMH levels are documented in the last three weeks before onset of estrus [15]. Surprisingly, recent data obtained from our research group using the same Canine AMH ELISA test kit (AL-116, Ansh Labs, Webster, TX, USA) in 18 healthy anestrous bitches found that AMH levels ranged between 0.22 and 22.24 ng/mL (mean ± SD, 5.95 ± 6.0) (unpublished data). The high variation in serum AMH levels in every stage of the estrous cycle between bitches [15] and in anestrus, as observed in the present study, suggested that future studies to establish a canine AMH reference range should include a larger number of animals and the stage of estrous cycle should be well-defined, i.e., early, mid or late anestrus. Furthermore, standardization of canine AMH ELISA tests must be developed. Our results indicated—as others did, too—that clinical application of AMH to monitor follicular development in periovulatory period in the dog is difficult because of large intra- and inter-individual variation. The effect of breed sizes on AMH concentrations was reported [15,23]; smaller breeds have higher AMH concentrations [23]. Because only Beagles were used, it seemed that the dog breed did not affect our current findings.

Interestingly, AMH concentrations did not differ significantly during the whole observation period, specifically from proestrus to the beginning of diestrus. This observation is in contrast with observations made by Walter and colleagues [15], who identified significant differences in circulating AMH between stages of the canine estrous cycle. The AMH levels decreased from early proestrus to preovulatory estrus and further declined from postovulatory estrus to metestrus [15]. It remains to be clarified whether this discrepancy might be related to differences in naturally occurring and medically induced estrus. Moreover, considering the timing of sample collection, blood samples were first collected on day 6 after estrus induction using deslorelin implants (approximately 6–8 days before ovulation) in the present study, while the two-fold increase in AMH levels in natural estrous cycles was detected between days 8 and 9 before the LH surge (approximately 10–11 days before ovulation) [14]. This might explain why no significant differences in AMH concentrations were detected in our study. To better understand the pattern of AMH and follicular development in induced estrus that potentially differ from natural-occurring estrus, further research should closely monitor AMH immediately after GnRH agonist administration.

The observation [36] that AMH and its transcribed gene expression were higher in proestrus and estrus compared to other stages of the reproductive cycle, but not significantly different between proestrus and estrus, fits our observation that no statistically significant difference in hormone concentrations could be detected after induction of estrus. However, as the first sample was taken during anestrus, a difference to subsequent AMH concentrations could have been expected. Some studies report smaller litter sizes following GnRH agonist-induced estrus compared to natural estrus [8,13], whereas others did not find any difference between natural and induced estrus in the same bitch and considered fertility to be unaffected [5,10]. Without doubt, follicle recruitment, growth and maturation differ between natural and deslorelin-induced estrus cycles, although details in deslorelin-induced estrus require further investigation. It is well known that bitches come into estrus ranging from 2 to 7 days after deslorelin administration [4,5,9,30], indicating that the transition from anestrus to proestrus, follicle recruitment and development is fast following treatment with the GnRH superagonist deslorelin. The mode of action is considered to be the increase in gonadotropins. Short-term administration of deslorelin has been shown to effectively increase LH concentrations, but it seems obvious that follicle-stimulating hormone (FSH) also increases as a consequence of the initial stimulatory effect. To our knowledge, details on serial FSH secretion following deslorelin treatment in bitches have not yet been published, but differences to natural FSH secretion might be an explanation for altered AMH patterns between natural and medically induced estrus in the bitch. Details about how AMH affects follicle recruitment and growth in the dog have not been performed yet and a model for estrus induction seems advantageous.

Until now, most studies investigating AMH effects and interactions between AMH and follicle development used mice and rats; further data are available in humans [20,37,38]. Despite certain agreements, these studies indicated obvious species-specific differences [39,40,41]. While AMH inhibits follicle growth via reduced granulosa cell proliferation in mice [37], AMH had a stimulatory effect on FSH-induced follicle growth in the rat [42]. Although a small number of animals was used in this study, our findings provided background for further investigation on AMH and follicle recruitment and growth in natural estrous bitches compared to medically induced estrous bitches using a deslorelin-containing slow-release implant and/or oral cabergoline. Further studies should also involve reproductive status, body weight and age of bitches.

In this study, ovarian ultrasonography was used to monitor the follicular changes after deslorelin implantation. The fluid-filled structure on the ovary was obviously identifiable in all bitches. The number of follicles was significantly higher on day 8 than the consecutive days. Decreases in the number of fluid-filled structures were observed near the time of ovulation in accordance with a previous study [43]. Moreover, the mean area of the ovaries significantly increased from day 8 to soon after ovulation (day 16). In a previous study, the ovaries gradually increased in size (length, width and depth) from proestrus to the day of ovulation. The explanation of increased ovarian size could be due to the increase in size of follicles [43] and the protrusion of growing follicles and/or newly formed corpus lutea [29].

## 5. Conclusions

Although a slow-release GnRH agonist (deslorelin) implant successfully induced estrus and ovulation in bitches, AMH concentrations and the course of AMH were different compared to earlier reports in natural estrus. It remains to be clarified whether the difference is related to estrus induction using GnRH agonist deslorelin and indicates the need for future studies comparing natural and medically induced estrus. Different to humans, AMH is not suitable to monitor ovarian response after deslorelin implantation in female dogs.

## Figures and Tables

**Figure 1 animals-13-00258-f001:**
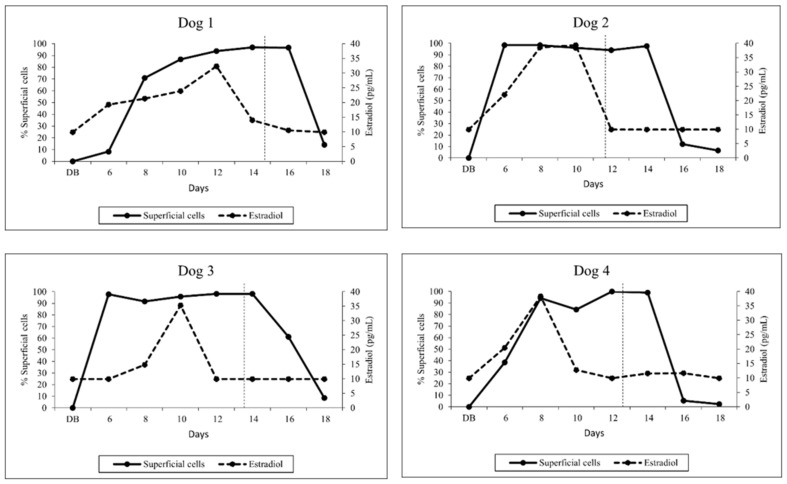
Individual percentage of superficial cells and serum estradiol concentrations of four dogs from the day before implantation (DB) to day 18 after implantation of a 4.7 mg deslorelin implant. The vertical dashed line indicates ovulation date.

**Figure 2 animals-13-00258-f002:**
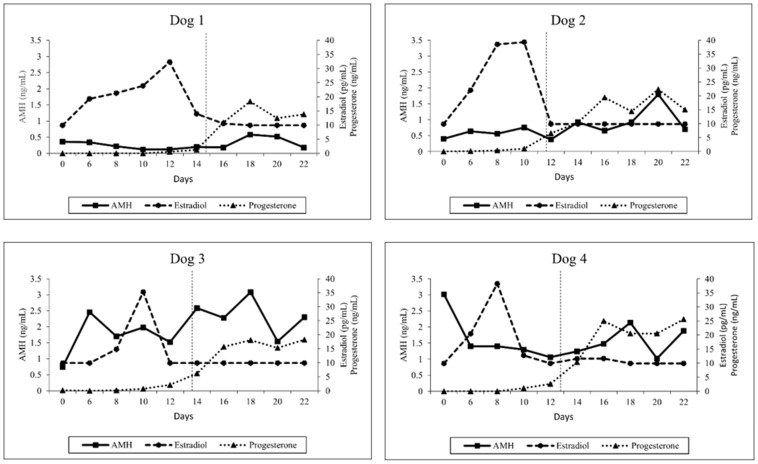
Individual serum progesterone, estradiol and AMH concentrations of four dogs from day 0 to day 22 following treatment with a 4.7 mg deslorelin implant. The vertical dashed line indicates ovulation date.

**Figure 3 animals-13-00258-f003:**
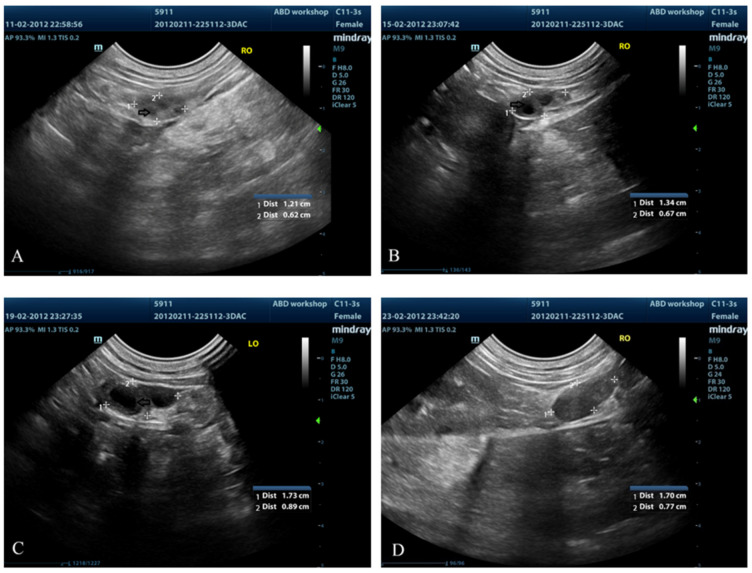
Examples of ovarian ultrasound images showing changes in size of the anechoic fluid-filled structures (open arrow) in a dog on day 6 (**A**), day 10 (**B**), day 14 (**C**) and day 18 (**D**) after treatment with a 4.7 mg deslorelin implant.

**Table 1 animals-13-00258-t001:** Mean (±SD) number of clearly identifiable fluid-filled structures on both ovaries and area of the ovaries from day 8 to day 18 after deslorelin treatment. Different superscripts of the mean number of follicles (^a^, ^b^) and the mean ovarian area (^A^, ^B^) differ significantly (*p* < 0.05).

Day	Mean Number of Follicles	Mean Ovarian Area (cm^2^)
8	7.25 ± 0.96 ^a^	0.74 ± 0.11 ^A^
10	3.50 ± 3.12 ^b^	0.76 ± 0.24 ^AB^
12	6.50 ± 6.36 ^b^	0.75 ± 0.23 ^A^
14	5.50 ± 4.95 ^b^	0.73 ± 0.24 ^A^
16	5.0 ± 3.0 ^ab^	0.80 ± 0.36 ^B^
18	4.0 ± 1.41 ^b^	0.83 ± 0.38 ^AB^

## Data Availability

Not applicable.

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
