# Peer review of "Serum Anti-Müllerian Hormone Levels and Estrous Monitoring of GnRH Agonist Deslorelin-Induced Estrus in Bitches: A Pilot Study"

_animals, 2023, doi:10.3390/ani13020258_

Round 1

Reviewer 1 Report

Authors aim to evaluate the effectiveness of AMH as a marker of follicular progression and fertility following desorelin treatment in female dogs as has been utilized for other species.  The grammar is relatively readable but it is difficult to track and state where corrections need to be made without line numbers for the manuscript.  Additionally, none of the figures reflect the results in the text.  Only one scatter plot line is apparent and it is at zero for all figures.  Figure should show the means for each hormone, not the individual hormone profile for each dog.  If statistical analysis is conducted then means and standard errors were calculated.  Therefore present the outcomes of the statistical analysis, i.e., means and standard error of the means that lead to the generation of the significant p values.  The manuscript is unacceptable in its current form due to the poorly constructed graphic data. Some additional comments below:

General comments:

1. Line numbers need to be added to the text

2. Figures need to be completely redone.  None reflect the text data in the results.

Summary

1.     Line 5: replace ‘involves in’ with ‘is involved in the’

2.     Line 8: as a comma before the word ‘or’

Abstract

1.     Line 8: replace ‘estrous’ with ‘estrus’.  Assuming authors meant stage of sexual receptivity rather than the entire reproductive cycle.

Section 2.1 Animals

1.     Line 1: delete ‘and’ after Beagle bitches.

Results:

1.      All figures show a single scatter plot line at zero over time.  All figure graphs must be replaced to reflect the results in the text.

2.     Additionally, authors should consider separate graphs that illustrate the means of AMH, estradiol, and progesterone over time for all 4 animals rather than each individual animal.  The reason for showing each individual animal is not valid.

3.     Figure titles are misaligned with figures.

Author Response

# Reviewer 1
1.Line numbers needed to be added to the text
With the version of manuscript provided by the journal, I tried to insert line numbers to the text but I could not. I do agree that, without line numbers, it is very difficult to track and state where corrections needed to be made.

2.Figures need to be completely redone.  None reflect the text data in the results.
The original manuscript contained incorrect Figures 1 and 2. It is most likely due to the file uploading process. The correct Figures have been included.  

3.Summary
Line 5: replace ‘involves in’ with ‘is involved in the’
This has been corrected.
Line 8: as a comma before the word ‘or’
This has been corrected.

4.Abstract
4.1 Line 8: replace ‘estrous’ with ‘estrus’.  Assuming authors meant stage of sexual receptivity rather than the entire reproductive cycle.
This has been corrected.

5.Section 2.1 Animals
5.1 Line 1: delete ‘and’ after Beagle bitches.
This has been corrected.

6.Results:
6.1 All figures show a single scatter plot line at zero over time.  All figure graphs must be replaced to reflect the results in the text.
The correct Figures have been included.  
6.2 Additionally, authors should consider separate graphs that illustrate the means of AMH, estradiol, and progesterone over time for all 4 animals rather than each individual animal.  The reason for showing each individual animal is not valid.
We find this suggestion is very useful. However, the mean values and standard deviation of AMH, progesterone and estradiol concentrations of 4 animals have been shown in the section 3.2 (Hormone analysis) in the original manuscript. Because there are a limited number of animals used in this study (as stated in Discussion), changes in hormone levels over the observation period were rather present individually (Figure 1 and 2). In this study, we believe that showing the data individually will give readers more information and clearly support our conclusion that there were large intra- and inter-individual variations on circulating AMH levels in the canine estrous cycle.
6.3 Figure titles are misaligned with figures.
The original manuscript contained incorrect Figures 1 and 2 including the figure titles, most likely due to uploading process. This has been corrected.

Reviewer 2 Report

The  small number of animals was used requires further studies

The paper is well written, clear and complete

Author Response

# Reviewer 2
1.The  small number of animals was used requires further studies
The paper is well written, clear and complete.
We agree with referee’s comment regarding the number of animals used in this study (as stated in Discussion section). However, all deslorelin-treated dogs showed estrous signs similar to natural occurring estrus. Although the number of animals were limited, serial blood samplings were collected before (Day 0) and after induction of estrous (Day 6, 8, 10, 12, 14, 16, 18, 20 and 22) resulting in 10 samples per animal. In addition, the effect of dog breed on AMH levels previously reported was fully aware, and only Beagle dogs were included to avoid breed differences in AMH concentration (as also stated in the original manuscript). We believed that monitoring of changes in hormonal profiles observed in this study provided fundamental evidence demonstrating that serum AMH concentrations were not differ significantly from proestrus to the beginning of diestrus in deslorelin-induced estrous bitches. Moreover, as suggested by another referee, “a pilot study” has been included in the title of our study.  

Reviewer 3 Report

Thank you for presenting your interesting research. The manuscript is well written, but you probably didn't realize you uploaded completely inappropriate graphs unfortunately (but this can be easily corrected). My main concern is, of course, really small number of included animals, from which any reall conclusions can not be adrequately drawn. Consider renaming it a "pilot study" if you are thinking about expanding of the research. But nevertheless, here are my main recommendations for revisions:

General/language:

There is a lot of unnecessary capitalization of words (e.g. Gonadotropin, Granulosa cells, Day 0, Serum Clot Activator etc). This should be corrected throughout the manuscript. There are also some other minor spelling and grammar mistakes (for example »estradio concentrations tented to decrease« etc), which should be coorected.

The "Simple summary" is far to scientific and technical. According to the manuscript instructions itshould be written for a lay audience, but I don't think that laypeople would understand "estrous induction", "follicular development", "hypogonadotropic anovulation" and many other similar terms. It should be significalntly signified, because in this form it doesn't differ from "normal abstract".

Introduction:

Following implant administration in anestrus, bitches come into estrus approximately 4 days (range 2-7 days). This sentance seems to be incomplete.

M&M:

2.1. Animals

Progesterone levels were below 1.0 ng/mL…. measured using which method?

2.2. Estrous induction

The implant was removed. How? Please briefly describe the method (because it can be a welfare issue for the dogs, I would like to see how it was done).

Results

Figure 1, Figure 2

Graphs don't seem to be the right ones, please provide the correct graphs.

Furthermore: the graphs shoud not duplicate the same information that is already provided in the text, which is probably the case here (I can't comment, because graphs are just flat-lines). Additionally, in figure 1 you already estradiol concentration in these dogs, but you put it again (probably) in Figure 2 – so, again, duplication of information. Please consider optimizing data presentation in graph (so it is not repetition of text and other graphs) and maybe putting all 4 graphs at least in a »collage«, so they are not 4 huge individual graphs, but more like 4 smalle graphs in 2x2 grid.

 Check the whole results section and consider optimization of presentation of results without duplication of the same information, the size and presentation of graphs etc.

3. Ovarian ultrasound: Replace it with Ultrasonografic examination of ovaries«

Figure 3. Examples of ovarian ultrasound images showing changes in fluid-
filled structures in a dog on Day 6 (A), Day 10 (B), Day 14 (C) and Day 18 (D)
after treatment with a 4.7 mg deslorelin implant.

It should eb mentoned what kind of changes? In size, in number, in composition,…? Also, the pictures should be equipped with arrows and explanatons what is what on the picture. Somebody, who is not very familiar with ultrasound, whouldn't know what on the picute is actually ovary.

Discussion

Major drawbacks of the strudy should be mentioned and discussed how the affected results.

Conclusions

Currently and different to humans, based on the results obtained in this study, AMH is not suitable as a diagnostic marker to monitor ovarian response after deslorelin implantation in female dogs…. This clonclusion should be changed. You can't use  »different to humans…. AMH is not suitable to monitor ovarian response after deslorelin implantation«, bevause deslorelin impant is not used in humans. Furthermore, the study was done on much too small number of dogs to draw any conclusions regrding generalization of results.

References:

There shouldn't be reference no. 44, because you have only 43 references in the text. There was some mis-numbering of the last reference in the list

Author Response

# Reviewer 3
1.There is a lot of unnecessary capitalization of words (e.g. Gonadotropin, Granulosa cells, Day 0, Serum Clot Activator etc). This should be corrected throughout the manuscript. There are also some other minor spelling and grammar mistakes (for example »estradio concentrations tented to decrease« etc), which should be corrected.
A: Thank you for the comments and these have been corrected in the revised manuscript.

2.The "Simple summary" is far to scientific and technical. According to the manuscript instructions it should be written for a lay audience, but I don't think that laypeople would understand "estrous induction", "follicular development", "hypogonadotropic anovulation" and many other similar terms. It should be significalntly signified, because in this form it doesn't differ from "normal abstract".
A: We find the referee’s suggestion is very useful. The simple summary has been revised. However, albeit within the scope of the journals, the objectives of our study were very specific to certain audiences who interests in the area of canine reproduction, it is difficult to avoid using some technical/ scientific words.  

3.Introduction:
Following implant administration in anestrus, bitches come into estrus approximately 4 days (range 2-7 days). This sentance seems to be incomplete.
A: This sentence has been revised.

4.M&M:
2.1. Animals
Progesterone levels were below 1.0 ng/mL…. measured using which method?
A: Using RIA or ELISA as published in previous publications.
2.2. Estrous induction
The implant was removed. How? Please briefly describe the method (because it can be a welfare issue for the dogs, I would like to see how it was done).
A: Under local anesthesia with 2 ml lidocain 2%, a small incision was made using a scalpel blade on the location of the implant and the implant was removed. References [4,5] have been included in the revised manuscript.
1. Fontaine et al. Induction of fertile oestrus in the bitch using Deslorelin, a GnRH agonist. Theriogenology 2011, 76, 1561-1566.
2. Walter et al. Estrus induction in Beagle bitches with the GnRH-agonist implant containing 4.7 mg Deslorelin. Theriogenology 2011, 75, 1125-1129.

5. Results
5.1 Figure 1, Figure 2 Graphs don't seem to be the right ones, please provide the correct graphs.
A: The original manuscript contained incorrect Figures 1 and 2. It is most likely due to the file uploading process. The correct Figures have been included. All figures are in the last section following the reference list in the revised manuscript.
5.2 Ovarian ultrasound: Replace it with Ultrasonografic examination of ovaries«
A: This has been revised in section 2.5.
5.3 Figure 3. Examples of ovarian ultrasound images showing changes in fluid-
filled structures in a dog on Day 6 (A), Day 10 (B), Day 14 (C) and Day 18 (D)
after treatment with a 4.7 mg deslorelin implant.
It should  e  entioned what kind of changes? In size, in number, in composition,…? Also, the pictures should be equipped with arrows and explanatons what is what on the picture. Somebody, who is not very familiar with ultrasound, whouldn’t know what on the picute is actually ovary.
A: This has been revised and the figure legend is corrected to read “Figure 3 Examples of ovarian ultrasound images showing changes in size of the anechoic fluid-filled structures (open arrow) in a dog on day 6 (A), day 10 (B), day 14 (C) and day 18 (D) after treatment with a 4.7 mg deslorelin implant.”

6. Discussion
6.1 Major drawbacks of the study should be mentioned and discussed how the affected results.
A: We find the referee’s suggestion is very useful. All major drawbacks of this study have been discussed including the limited number of animals used in this study (moreover, as suggested by the referee, “a pilot study” has been included in the title of our study), lower levels of AMH in comparison to previous studies, variation in AMH levels among studies and closely monitoring AMH immediately after GnRH agonist implantation in further research.
7. Conclusions
Currently and different to humans, based on the results obtained in this study, AMH is not suitable as a diagnostic marker to monitor ovarian response after deslorelin implantation in female dogs…. This clonclusion should be changed. You can't use  »different to humans…. AMH is not suitable to monitor ovarian response after deslorelin implantation«, bevause deslorelin impant is not used in humans. Furthermore, the study was done on much too small number of dogs to draw any conclusions regrding generalization of results.
A: We do agree with the referee's suggestion and the statement has been revised.

8. References:
There shouldn't be reference no. 44, because you have only 43 references in the text. There was some mis-numbering of the last reference in the list
A: We apologize for the mistake. Reference No. 44 is now deleted.

Round 2

Reviewer 1 Report

Authors have adequately addressed the concerns and comments of the reviewer.

Reviewer 3 Report

Thank you for taking into account my suggestions and improving the manuscript.